# g4ppyy: automated Python bindings for GEANT4

**P. Stowell⋆, R. Foster and A. Elhamri**

School of Mathematical and Physical Sciences, University of Sheffield, S10 2TN, UK

⋆ p.stowell@sheffield.ac.uk

## Abstract

GEANT4 is a particle physics simulation tool used to develop and optimize radiation detectors. While C++-based examples exist, Python's growing popularity necessitates the development of a more accessible Python bindings interface. This work demonstrates the use of cppyy, the automated C++-Python binding package, to provide an accessible interface for developing applications with GEANT4. Coupled with newly developed Python visualization tools and a Python-specific helper layer, we demonstrate the suitability of the interface for use in constructing simplistic simulation scenarios showing some initial benchmarking studies when compared to a pure C++ equivalent simulation example.

# 1 Introduction

GEANT4 [1–3] is a sophisticated high energy particle physics simulation framework used to develop and optimize radiation detectors. Including a detailed library of different physics processes and configurable physics lists, the framework has become the common standard for simulating radiation detectors in the high energy physics community and also has widespread use in other applied areas such as the medical physics, nuclear physics, and space radiation modeling sectors.

Written in C++, the framework benefits from a substantial collection of examples to support novice users, and several courses are available focused on introducing the core concepts to new users. Given the hierarchical structure of the code-base a major requirement for development in GEANT4 beyond very simple use cases is a good understanding of C++ class inheritance. This means use of the code by new users with limited coding experience can be difficult as C++ development and build procedures must be first be understood before considering the core simulation principles GEANT4 itself. This particularly limits its use among undergraduate students, with many university physics courses beginning to adopt Python as a standard programming language over C++.

Whilst Python bindings based on pybind11 [4] have been developed in the past, such as g4python [5], or geant4-pybind [6], these have been limited to selected parts of the entire GEANT4 framework due to limitations need to develop wrapper interfaces in pybind around every purely virtual class within GEANT4. For a Python binding of a large framework to be effective, ideally several usage constraints need to be satisfied at the same time. These are namely:

1. Coverage : the implementation of bindings with as much coverage as possible of the original codebase,

2. Mapping : direct mapping where possible of the original structure so that the large collection of C++ focused tutorial materials can still be used as a guiding basis,

3. Efficiency : comparable performances between the C++ and Python interfaces within an order of magnitude, recognizing that for simple use cases there is a trade-off between CPU time and initial development time,

4. Visualization : ability to produce visualizations of GEANT4 geometries consistent with the common interactive approaches available when running GEANT4 natively,

5. Containerization : ability for the code to be setup and developed by new users within containerized environments, preferably accessible through web based development environments such as Jupyter [7].

The last requirement on containerization is directly correlated with visualization requirements, as to date GEANT4 does not have an official web-based interactive visualization interface. In simple examples of linear accelerators in g4python the use of GEANT4's internal ray tracer or DAWN file outputs have been demonstrated to produce static images of loaded geometries [8], but lack the ability to navigate a scene and understand the full geometry hierarchy in full.

In this work we demonstrate the development of an automated Python binding loader, g4ppyy[1], which is focused on the use of GEANT4 in a containerized Jupyter environment with the cppyy package [9,10]. cppyy is an automatic Python to C++ binding tool built upon the Cling [11] dynamic interactive C++ interpreter. Capable of dynamically loading C++ libraries,

---

[1]The source code is available at github.com/patrickstowell/g4ppyy.

and just-in-time compilation of C++ definitions directly in Python, the package provides a way to automatically generate a complete set of efficient bindings for a large framework such as GEANT4. `cppyy` was originally developed to support similar problems with the ROOT high energy physics analysis framework, acting as the backend for PyROOT [12].

This article describes the principles behind each of the g4ppyy components and demonstrates their application. Section 2 gives a brief overview of the GEANT4 interface and the core concepts usually covered in the most basic GEANT4 C++ examples. Section 3 describes the lazy loading binding interface implemented in g4ppyy and gives examples on how this can be used to access the entire capabilities of the GEANT4 framework at run time, Section 4 describes the Python-based visualization tools in g4ppyy, Section 5 demonstrates additional helper interfaces available in g4ppyy to support rapid development of simulation problems, and finally Section 6 gives a summary of the capabilities and areas for future development.

## 2   GEANT4 overview

GEANT4 is used to simulate the passage of particles through matter and is flexible enough to support simulations in a wide range of radiation physics applications. Whilst containing a vast number of problem specific components, every simulation in GEANT4 has a common set of required components that must be defined by an application developer which are described below.

### 2.1   Geometry

The user must provide a description of the geometry that they wish to propagate particles through. The geometry may be as simple as defining a small collection of primitive boxes or could include thousands of geometrical elements that make up the complex geometry of a detector at a collider experiment. It is mandatory for a user to provide a description of the geometry by deriving a class from the `G4VUserDetectorConstruction` base class before defining the world geometry within a `Construct()` function.

The description of a geometry is split into three components: solids (`G4VSolid` and its subclasses), logical volumes (`G4LogicalVolume`), and physical volumes (`G4VPhysicalVolume` and its subclasses). Solids describe the shape of the geometry. GEANT4 provides a series of constructive solid geometry (CSG) primitives, such as boxes, tubes, spheres, etc, that may be combined using boolean operations to create more complex geometries. Also included are tesselated solids, which are useful when importing arbitrary geometrical shapes from CAD software. Logical volumes are instances of solids that describe how the geometry should behave in the simulation. Firstly, it allows a material to be assigned to a solid which will govern how particles will interact with the geometry. Secondly, it acts as a key element for nesting geometries to create a hierarchical geometry tree. Each logical volume is aware of its daughter volumes and how they are placed relative to one another. Physical volumes are instances of logical volumes that have been spatially position within the world. Physical volumes range from the straightforward `G4PVPlacement`, which positions the volume at a user-specified position and rotation relative to its mother volume, to more complex `G4PVReplica` which places several instances of a logical volume using some user-defined pattern.

### 2.2   Physics lists

GEANT4 offers a series of libraries for modeling particle interactions across many energy ranges. Since not all interactions are relevant for all scenarios being simulated, the user is able to customize which physics lists are activated for a simulation. Whilst there is support for users to

define their own physics lists or even individual particle processes, the majority of use cases involve the loading one of GEANT4's prepackaged physics lists which consistent handling of different fundamental physics across a specific energy scale.

## 2.3 Primary particle generation

Each event in the simulation begins with the generation of one or more primary particles, the types and properties of which are specified by the user. The user may use a generator provided by GEANT4 or define their own. Two of the most commonly used internal generators are the `G4ParticleGun` and the `G4GeneralParticleSource`, both of which allow for the user to specify particle properties, such as the type of particle, it's position, momentum direction, energy, etc. The `G4GeneralParticleSource` allows for more complex configurations of primary particle generation, such as generating particles with varying position, direction, or energy distributions. When defining a custom generator it is mandatory for the user to create a class derived from the `G4VUserPrimaryGeneratorAction` base class and to implement the `GeneratePrimaries` method which is called at the beginning of each event to create the primary particles.

## 2.4 The simulation hierarchy

The GEANT4 simulation loop can be thought of as being split into several simulation units organized into a hierarchy. The largest unit is called a `Run`, which consists of many `Events`. An `Event` may consist of many `Tracks` representing particles traveling throughout the world. A `Track` consists of many `Steps`, which are the smallest unit of simulation and correspond to the positions and properties a particle has as it propagates – or 'steps' – through the simulated world. At each level of the simulation hierarchy, there are action classes which implement callback functions (`G4UserRunAction`, `G4UserEventAction`, etc) that are provided to allow the user to execute custom code within the simulation loop itself. For example, a user may define a custom stepping action to measure the energy deposition of a particle in a volume as it travels through. The user stepping action gives the user access to the current particle step and the information contained within it, such as the energy deposition within that step.

One common requirement within radiation detector simulations is the need to understand particle behavior, for example energy deposition, as particles enter specific logical volumes. To achieve this, volumes within the simulation can be defined to be sensitive detectors, which use the step information of particles traversing the detector volume to create `hits`. A `hit` is a container for user-specified information about the particle that entered the volume such as position, time, momentum, energy deposition, etc. Sensitive detectors can filter when they record information by a user-provided method which can select only certain particle types, energy ranges, etc. These hits can then be collected for all registered sensitive detectors by custom event actions.

User-defined actions are one of the common ways to export information from the simulation, and any Python based interface must therefore also allow users to define custom inherited classes for the main action types that can be added directly into the simulation loop in the same way as this is handled within C++. A schematic of the hierarchy is shown in Figure 1.

## 3  `import g4ppyy`

cppyy provides a standard interface for loading external functions or classes from dynamically linked libraries by adding them to an global namespace , e.g. `cppyy.gbl.G4Box(...)`. All

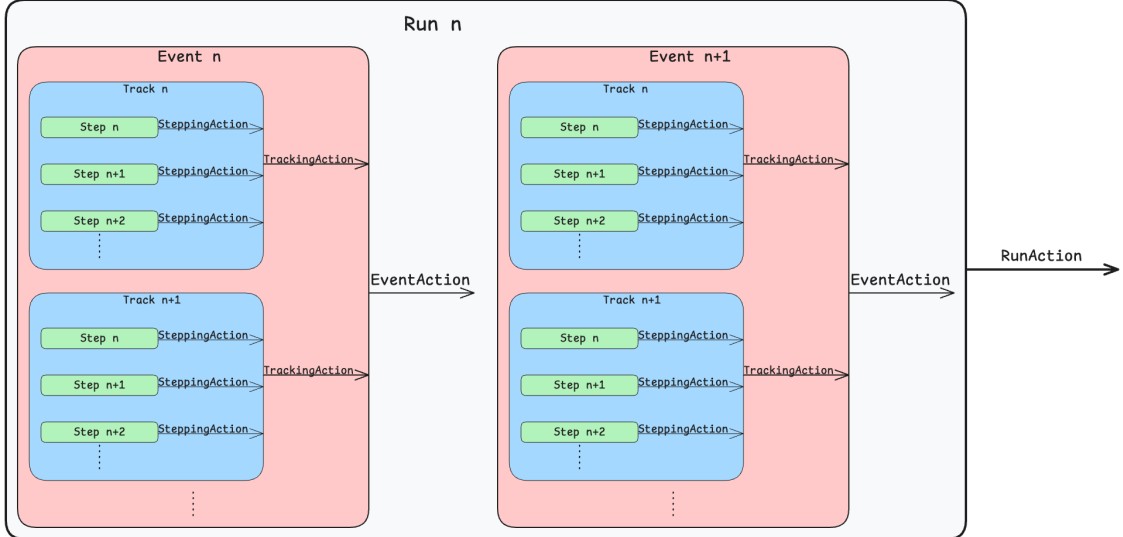

Figure 1: Schematic showing a simplified version of the simulation hierarchy in GEANT4. Runs contain many Events which contain many Tracks which contain many Steps. Arrows denote function invocations by each action class.

the GEANT4 classes exist in a single namespace. To make it easier for new users to focus on specific aspects of the framework, g4ppyy acts as an additional loading wrapper around cppyy for GEANT4 4. On import g4ppyy first queries the user's environment to determine the locations of the standard GEANT4 headers and libraries. This is then used to dynamically load all required pre-compiled libraries for GEANT4, and register the path for known includes. The importing of headers is limited at the start to a collection of base classes that are either required for all simulations (e.g. `G4RunManager`), or used throughout the codebase (e.g. `G4ThreeVector`, SI unit definitions). After which a series of Python specific helper functions are added to the g4ppyy module itself as an extra layer which are detailed specifically in Section 5. Following the first import, g4ppyy provides an efficient lazy loading system that automatically searches the GEANT4 include paths for relevant GEANT4 definitions and adds the corresponding header files only at first use. Relying on cppyy allows the development of GEANT4 applications as the bindings generated can accommodate the class inheritance structure required for many applications automatically. As shown below it is possible to define custom detector construction classes inherited from `G4VUserDetectorConstruction` within Python, before registering these with the run manager. The example below compares the definition of this in Python to the C++ equivalent.

## 4  Python visualization

A major consideration when developing tools for new users is the ability to both visualize the geometry, underlying physics processes, and final simulated data outputs in an efficient manner. The development of a consistent world geometry is a necessity to obtain sensible results, and significant time can be wasted trying to debug geometry misalignment or overlaps. The OpenGL-based QT interface for GEANT4 is widely used to check geometries, providing efficient ways to navigate the world, and draw individual particle trajectories. Alternative interfaces are also available based on external requirements (DAWN, VRML, OpenInventor), or even GEANT4's internal ray tracing code.

The majority of the existing visualization solutions require some level of machine spe-

```cpp
class DetectorConstruction : public G4VUserDetectorConstruction {
public:
  inline G4VPhysicalVolume* Construct(){
    G4NistManager* nist = G4NistManager::Instance();

    G4Material* boxMaterial = \
        nist->FindOrBuildMaterial("G4_WATER");

    G4Box* solidBox = new G4Box("Box", 5*cm, 5*cm, 5*cm);

    G4LogicalVolume* logicalBox = \
        new G4LogicalVolume(solidBox, // Solid
        boxMaterial,  // G4Material
        "Box"); // Name

    G4VPhysicalVolume* physicalBox = \
      new G4PVPlacement(0, // no rotation
        G4ThreeVector(),    // at (0,0,0)
        logicalBox,         // logical
        "Box",              // name
        0,                  // mother volume
        false,              // boolean operation
        0,                  // copy number
        true);              // check overlaps

    return physicalBox;
  }
};
```

Listing 1: Simplified example demonstrating the construction of a geometry consisting of a single box using C++.

```python
import g4ppyy as g4

class CustomDetectorConstruction(g4.G4VUserDetectorConstruction):

    def Construct(self):
        nist = g4.gNistManager

        box_material = nist.FindOrBuildMaterial("G4_WATER")

        cm = g4.SI.cm # SI units wrapped in namespace
        solid_box = g4.G4Box("Box", 5*cm, 5*cm, 5*cm)

        logical_box = g4.G4LogicalVolume(solid_box, # Solid
            box_material,  #G4Material
            "Box")         #Name

        physical_box = g4.G4PVPlacement(0,       # no rotation
                        g4.G4ThreeVector(), # at (0,0,0)
                        logical_box,        # logical
                        "Box",              # name
                        0,                  # mother
                        False,              # boolean
                        0,                  # copy number
                        True)               # check overlap

        return physical_box
```

Listing 2: The equivalent example using g4ppyy Python bindings.

cific setup by the user. Ideally any solution for use in tutorials for novices needs to minimize setup of these external requirements. To overcome this challenge, g4ppyy implements two Python-based GEANT4 visualization managers built upon well established Python plotting tools, namely Matplotlib [13], and Jupyter-K3D. Matplotlib is a Python graphing package that is widely used to make publication-ready plots from analysis of data objects and is capable of 2D and 3D plotting. K3D is a three dimensional graphing tool that leverages the webGL plotting API available in most modern web browsers to construct interactive 3D images that can be navigated with high performance using GPU acceleration on the host machine. Both are readily available Python packages that have minimal external dependencies or setup requirements, and can be used to draw images directly within web-browser based Jupyter development environments.

These g4ppyy visualization managers are built upon GEANT4's internal scene handler base classes on the Python side, making it easy for them to be used as a template for users to develop their own visualizers by simply overriding the corresponding primitive drawing functions. Drawing options are obtained directly from GEANT4's `G4Visible` attributes for each object's corresponding primitive. This means each visualizer is fully compatible with GEANT4's existing visualization drawing macros, allowing for drawing of filtering of particles based on their properties. The K3DJupyter interface produces a 3D OpenGL interactive interface which is also added directly to the output of a Jupyter cell. This comes with a widget panel that can be used to navigate the geometry and enable/disable specific components in the same way as the native GEANT4 OpenGL interface. The MPLJupyter provides interactive 2D projections in x-y, y-z, x-z planes for the entire geometry when called. Both visualization manager outputs are automatically added to the cell outputs when called within a Jupyter environment. Figure Figure 2 shows screenshots of these interfaces working in Jupyter.

## 5 Benchmarking

It is accepted that running GEANT4 through Python bindings will incur a performance penalty. However, the different components of the simulation will not all have the same impact on performance. For example, in the majority of simulations, the geometry will be constructed once before any events are simulated and then will never be changed during the course of the simulation. As a result, the performance impact of the Python layer relative to the overall execution time will be minimal compared to a native C++ implementation. The key area where performance will be a concern is in the action classes which are hooked in at various levels of the simulation hierarchy. The `SteppingAction` for example, may be invoked a multitude of times for a single particle track and if implemented through the Python bindings will result in the Python function code being interpreted an equal number of times. The same applies for any `SensitiveDetectors` and this will obviously incur a more serious performance penalty compared to C++. For beginner users, who are often running more simple simulations, these performance issues are unlikely to be a major concern. For more advanced users, there is the possibility for using interpreted Python action classes when setting up and prototyping the simulation and then using JIT compilation or hooking in compiled C++ classes once the desired behavior of the action class has been achieved.

Here we report on initial benchmarks measuring the performance of g4ppyy relative to the native C++ implementation of GEANT4. To benchmark the code, we adapt the commonly used example B1, from the basic library of GEANT4 examples provided by the authors. The geometry of the simulation, as can be seen in Figure 3, consists of a cone and a trapezoid inside a box of water with the remaining world volume filled with air.

Example B1, as provided in the GEANT4 software, contains several different macros to

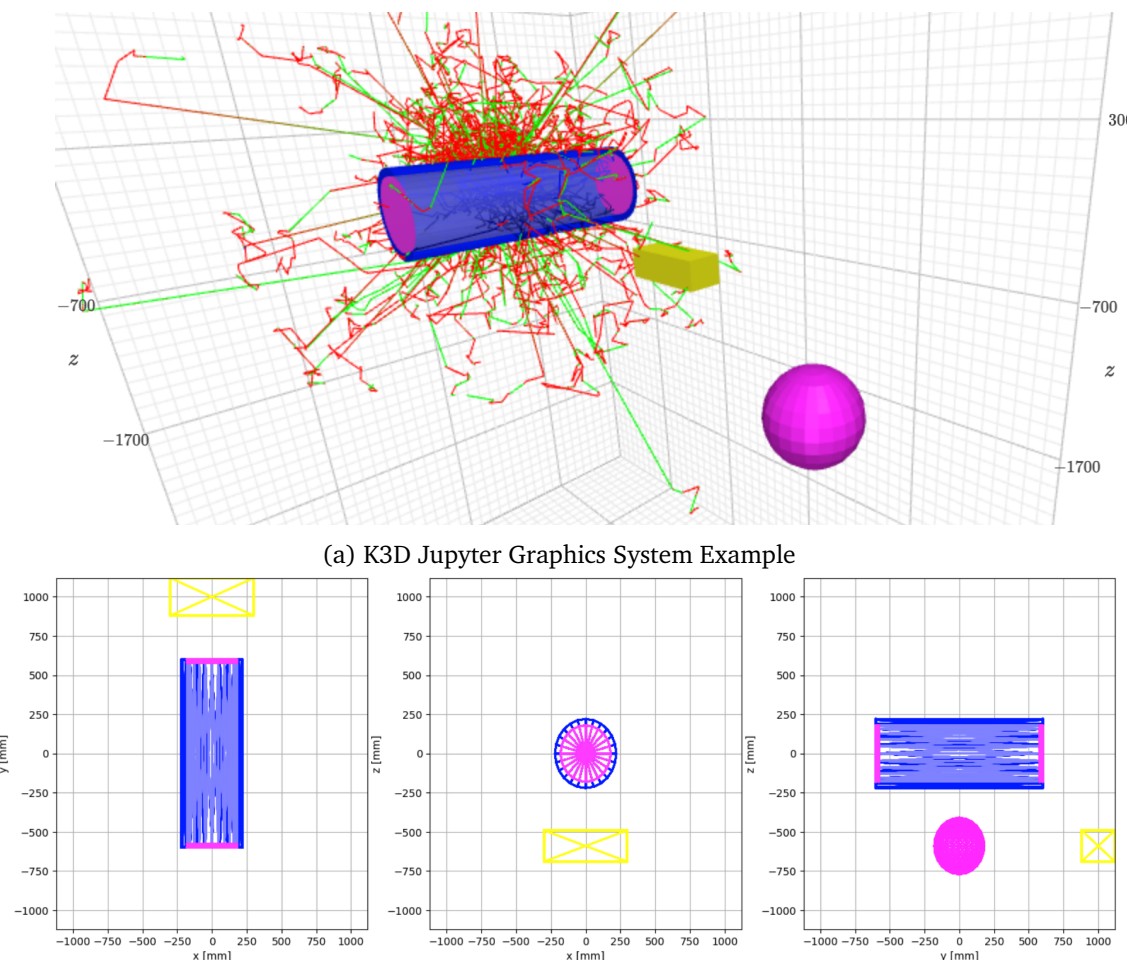

(a) K3D Jupyter Graphics System Example

(b) Matplotlib Jupyter Graphics System Example

Figure 2: g4ppyy K3DJupyter (a) and Matplotlib (b) interfaces. A detector consisting of a high-density polyethylene shield surrounding a Gd-loaded water Cherenkov detector is implemented (shown in blue and pink) and exposed to a beam of high energy neutrons (trajectories shown in red in the K3D image).

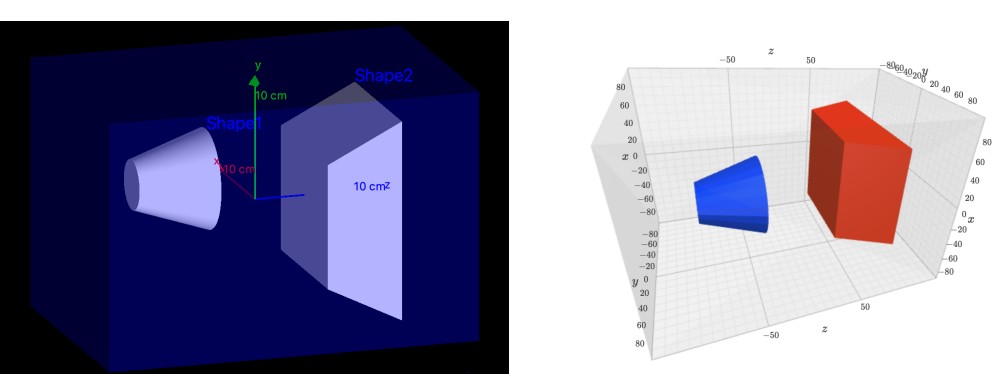

Figure 3: (Left) The example B1 geometry as observed through the GEANT4 interactive mode using OpenGL as the visualization driver. The green lines show a gamma particle that has been simulated. (Right) The equivalent geometry handled in g4ppyy through the K3D graphics system in a browser.

```
1 class SteppingAction : public G4UserSteppingAction
2 {
3 public:
4   inline void UserSteppingAction(){
5     if (!fScoringVolume)
6     {
7       const auto detConstruction = static_cast<const
    DetectorConstruction *>(
8         G4RunManager::GetRunManager()->GetUserDetectorConstruction()
    );
9       fScoringVolume = detConstruction->GetScoringVolume();
10    }
11
12    // get volume of the current step
13    G4LogicalVolume *volume =
14        step->GetPreStepPoint()->GetTouchableHandle()->GetVolume()->
    GetLogicalVolume();
15
16    // check if we are in scoring volume
17    if (volume != fScoringVolume)
18      return;
19
20    // collect energy deposited in this step
21    G4double edepStep = step->GetTotalEnergyDeposit();
22    fEventAction->AddEdep(edepStep);
23  }
24 };
```

Listing 3: The user stepping action as originally implemented in example B1.

generate different primary particles. In our case, we only simulate 6 MeV gamma particles that are produced at a random point on a plane on the left side of the water box (as seen from Figure 3) and travel towards the trapezoid. The stepping action is used to check at every step of the propagation of the gamma particle whether the particle is current in the trapezoid. If the gamma particle is in the trapezoid, then the energy deposition within that step is recorded and accumulated. At the end of each event, the energy deposition from all of the steps within the trapezoid is accumulated and at the end of the run the energy deposition from all events is totaled. This means that this benchmark features the main simulation components discussed in Section 2.

A Python version of example B1 was written using g4ppyy remaining as consistent as possible with the C++ reference. An example of the conversion of the UserSteppingAction along with the original C++ code is shown below. Validations with a fixed seed in both the C++ and Python equivalent found the measured energy deposition within the trapezoid at the end of a run to be identical in both the Python and C++ implementations. We also consider a third implementation, in which the stepping, event, and run actions are written in C++ and JIT compiled by cppyy before the simulation begins, but otherwise the setup is the same as the Python implementation. This intermediate implementation attempts to move some of the most commonly invoked functions from Python to compiled C++ to understand the performance impact incurred by repeated bridging of the Python-C++ interface.

The benchmarks were performed by varying the number of gamma particles simulated from 100 to 10,000,000 for each of the three implementations and measuring the execution time. Note that the measured time is the total time taken for the shell command which invokes the simulation and as such includes all pre-simulation steps such as the geometry construction. All measurements were made using the same PC and with the GEANT4 kernel running in single-

```python
class CustomSteppingAction(g4.G4UserSteppingAction):

    def __init__(self, event_action):
        super().__init__()
        self.event_action = event_action
        self.scoring_volume = None

    def UserSteppingAction(self, step):
        if not self.scoring_volume:
            run_manager = g4.G4RunManager.GetRunManager()
            det_con = run_manager.GetUserDetectorConstruction()

            self.scoring_volume = det_con.GetScoringVolume()

        prestep = step.GetPreStepPoint().GetTouchableHandle()

        volume = prestep.GetVolume().GetLogicalVolume())

        if volume != self.scoring_volume:
            return

        edep_step = step.GetTotalEnergyDeposit()
        self.event_action.AddEdep(edep_step)
```

Listing 4: The stepping action translated into Python.

threaded mode.

The benchmarking results are shown in Figure 4. As is expected, the C++ implementation is fastest. The intermediate implementation is a factor of between 2 and 7 times slower depending on the number of primary particles being simulated and the purely Python implementation is between 7 and 9 times slower. It is worth noting that simply importing the g4ppyy module, and the subsequent loading of C++ header files that the import initiates, takes approximately 3.5 seconds which explains a large proportion of the slower execution time for smaller numbers of primary particles simulated.

As the number of primary particles that are simulated increases, the application spends a larger proportion of the total execution time invoking the stepping action. For example, at 1,000,000 primary particles simulated, the Python implementation spends over half of its execution time (54%) in the stepping action compared to just 12% at 10,000 events. As a result, when the number of primary particles simulated increases, we see the performance of the Python implementation degrade relative to the C++ implementation. However, for the intermediate implementation with all action classes written in C++, we see very large improvements in performance relative to the pure Python implementation. Performance of the intermediate implementation will obviously never reach the C++ implementation as the commonly invoked primary generator action is still interpreted (but could be compiled in the same way as the action classes), and there will always be some overhead from bridging language domains.

We judge the relative performance of g4ppyy to be acceptable for a number of reasons. Firstly, execution time remains within an order of magnitude of the native implementation for all tests presented here, which is likely to be sufficient for many use cases. Secondly, development time is greatly improved for the many users who are more comfortable coding in Python rather than in C++. Entire simulations can be self-contained within a single Python file, rather than spread across many C++ source and header files as is demonstrated in the GEANT4 examples. This, in conjunction with the removal of the compile step, allows for rapid iteration and experimentation which is vital for new users to develop understanding. This makes GEANT4

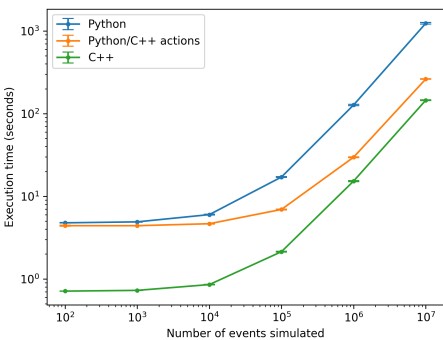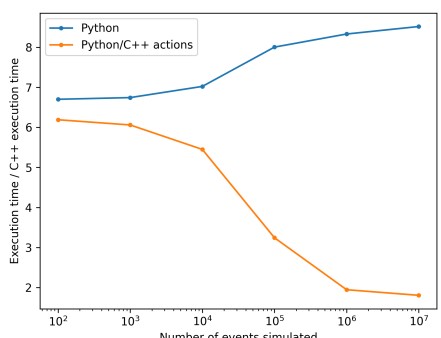

Figure 4: Performance benchmarks of g4ppyy using example B1. (Left) The execution times for varying numbers of simulated primary particles for the pure Python, intermediate, and native C++ implementations. (Right) The ratio of the Python and intermediate implementation execution times to the native C++ execution time.

simulations accessible to users who may not have even attempted to create their own simulations when required to work within a complex C++ framework. Thirdly, if performance is important then it is possible for more experienced users to achieve significant performance gains without significant rewrites by targeting commonly invoked functions to be moved to compiled C++. This approach still retains the flexibility offered by Python since any C++ can easily be defined and JIT compiled by cppyy within the same Python file that contains the rest of the simulation. We also envisage future developments that will improve performance while reducing the need to cross language domains, principally the addition of Numba support, which will allow for JIT compilation of Python functions to improve performance with minimal code rewrites.

# 6 Python helper layer

In addition to the loading helpers and visualization tools, g4ppyy includes a series of helper functions to make the creation of geometries and assignment of object attributes, such as those necessary for accurate optical tracking simulations, simpler for new users. A series of helper build commands are added that allow overloading of specific inputs similar to other Python APIs. These interfaces allow the use of tab-completion to be used to query inputs. The assumption for the geometry-focused helpers is that users will want to create single objects and place these directly into the world geometry at run time in a single command so that they can explore simple world constructions before touching on more complex concepts such as replica, cloned, or parametrized placements of logical volumes. Shown below are some of the example helper functions which support creation of custom materials, and direct creation and placement of logical volumes into a world geometry in a single command.

These helper functions combined with the easily override-able detector functions makes it possible to produce a complete simulation problem in a single short script with g4ppyy which can be iterated on when starting to develop more complex simulation scenarios.

```
1  # Make a new compound material from NIST and add optical data
2  loaded_water = g4.helper.build_material(
3          name = "LoadedWater",
4          density = 1*g/cm3,
5          materials = ["G4_WATER","G4_GADOLINIUM_OXYSULFIDE"],
6          fractions = [0.998,0.002],      # Mass Fractions
7          RINDEX_X = [0.1*eV, 3*eV, 10*eV] # Refractive Index Energies
8          RINDEX_Y = [1.33, 1.33, 1.33])  # Refractive Index Values
```

Listing 5: Python helper functions to support rapid prototyping of materials including their optical properties.

```
1   # Build a G4Box for the world made from loaded_water
2   world = g4.builder.build_component(
3       name = "world",
4       solid = "box",
5       x=4*m, y=4*m, z=2*m,       # Box Dimensions
6       material = loaded_water) # Box Material
7
8   # Build a G4Tubs POLYETHYLENE shell and place inside the world
9   hdpe_shell = g4.helper.build_component(
10      name = "hdpe_shell",
11      solid = "tubs",                # Tube Geometry
12      rmax=11*cm, z=0.6*m/2,         # Tube Dimensions
13      material = "G4_POLYETHYLENE", # Tube Material
14      mother = world,               # Mother Placement
15      rot = [90*deg, 0.0, 0.0],     # Rotation Angles
16      color = [0.0,0.0,1.0,0.8],    # Drawing options
17      drawstyle = "solid")          # Drawing options
```

Listing 6: Python helper functions to support rapid placement of logical volumes based on simple primitives.

# 7 Conclusion

We have demonstrated the potential for using cppyy to build automated C++-Python bindings for GEANT4. The wrapper package, g4ppyy, comes with a series of helper functions that allow users to focus on only discrete components of the GEANT4 framework, and allows the visualization of the world geometry and trajectories within an accessible web interface with minimal setup. Initial development examples demonstrate the suitability of using this package to build simple simulation problems, and benchmarking has demonstrated it is capable of performing basic simulations within a factor of 10 of that of the the C++ equivalent. Whilst this is expected to be slower for more complex simulation problems, it demonstrates the suitability of the bindings for use by novice users who may have limited C++ experience to learn the core principles of GEANT4. In addition we have demonstrated that the use of JIT compiled C++ bindings for the most CPU intensive components has a significant improvement compared to a full Python-based simulation routine. Future efforts are focused on development of extra helper functions to support rapid prototyping of detectors, and the potential for using acceleration structures to carry out JIT compilation of CPU intensive Python-side processing code to produce substantially more efficient simulations.

# Acknowledgments

**Author contributions**   P. Stowell: Core framework development and concept. R. Foster: Framework contributions, code testing, benchmarking, paper co-writing. A. Elhamri: Pre-alpha code testing and feedback.

**Funding information**   P. Stowell and R. Foster acknowledge funding from the STFC Consolidated Grant, STFC Nuclear Security Council, and EPSRC and STFC Impact Acceleration Award Accounts.

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
