# Peer review of "g4ppyy: automated Python bindings for GEANT4"

_SciPost Physics Codebases_

## Round 1 · Referee Report · Anonymous (Referee 1) · 2025-3-20

Strengths

This paper is an easy read, the central point is well argued, and detailed examples are provided. (The code described is open source and available on github, with clear installation instructions and further examples.) The authors clearly contextualize their work within the larger software ecosystem and community of users.

Weaknesses

Benchmarks are limited to CPU only, with no information on memory overheads. There are claims of a faster debug cycle, but only because their setup is simpler, no wall clock times given even as their approach adds several seconds to the startup time. The code describes only working examples, whereas the a claim of an improved development cycle should also consider the cost of debugging in a multi-language environment.

Report

A couple of questions came to mind when reading the paper. It would be
great if answers to these could be addressed (or otherwise resolved in
future work if they are out-of-scope for this paper, but in that case
they may still be worth mentioning as "out-of-scope").

p5: Searching include paths for relevant Geant4 definitions is something
that would only work if the code base is organized the way that Geant4
is and may not be generally applicable. Has there been an attempt to use
pre-compilation instead? If yes, what would the typical use/performance
differences be? (Noting here the one-off 3.5s mentioned on p10.)

Loading headers lazily also changes the loading order of definitions
based on use, i.e. from one application to the next. This may matter if
e.g. templates are used. Have problems been observed with this approach?

p7: "then using JIT compilation" ... It appears that this only talks about
including C++ code as text to be JITed? Numba is mentioned, but not
tried? The question comes up b/c Numba does not provide C++ support and
Numba support in cppyy is rudimentary. Additionally, if the callback is
through a derived class (as opposed to a function callback), there will
be a dispatcher class in between, which goes through Python and hence
locks the GIL. Is this a concern? (Noting that the paper only tested
single-threaded mode.) Different dispatchers that install callbacks
through function pointers instead of Python callables could be used.

p9: Listing 3 could do with manual line breaks for cleaner looking code
(e.g. the cast closing bracket being on a new line is a bit ugly).

"remaining as consistent as possible with the C++ reference" would
arguably mean "if self.scoring_volume is None:", not the boolean
conversion of "self.scoring_volume" as shown.

p11: Figure 4, left hand. It appears that the ratio between pure Python and
pure C++ remains constant, but the hybrid version shows relative
improvement. It also stays flat for longer. This suggest that the
initial overhead is larger than the per-event overhead for longer,
which seems reasonable enough? However, if so, why does it start out
lower than the pure Python version, i.e. suggesting lower initial
overhead? Was part of the geometry building also JITed?

Requested changes

Grammar:

p2: must be first be understood -> must first be understood
simulation principles GEANT4 -> simulation principles of GEANT4
due to limitations need to develop -> due to the need to develop
p3: been spatially position within -> been spatially positioned within
p4: for example energy deposition -> for example energy depositions
p5: GEANT4 4. -> GEANT4.
p7: (?) The MPLJupyter -> MPL Jupyter
Figure Figure 2 -> Figure 2
p9: particle is current in the -> particle is currently in the

As a style item (various pages), suggest "etc" -> "etc.".

Recommendation

Ask for minor revision

---

## Editorial Decision

awaiting_resubmission